# Effect of Paddy-Upland Rotation System on the Net Greenhouse Gas Balance as the Sum of Methane and Nitrous Oxide Emissions and Soil Carbon Storage: A Case in Western Japan

**Hiroyuki Hasukawa [1,\*], Yumi Inoda [1], Satoshi Toritsuka [1], Shigeto Sudo [2], Noriko Oura [2], Tomohito Sano [2,3], Yasuhito Shirato [2] and Junta Yanai [4]**

[1]  Agricultural Technology Promotion Center of Shiga Prefecture, Omihachiman 521-1301, Japan; inoda-yumi@pref.shiga.lg.jp (Y.I.); toritsuka-satoshi@pref.shiga.lg.jp (S.T.)
[2]  NARO Institute for Agro-Environmental Science, Tsukuba 305-8604, Japan; ssudo@affrc.go.jp (S.S.); nori@affrc.go.jp (N.O.); sanotomo@affrc.go.jp (T.S.); yshirato@affrc.go.jp (Y.S.)
[3]  NARO Headquarters, Tsukuba 305-8517, Japan
[4]  Graduate School of Life and Environmental Sciences, Kyoto Prefectural University, Kyoto 606-8522, Japan; yanai@kpu.ac.jp
\*  Correspondence: hasukawa-hiroyuki@pref.shiga.lg.jp; Tel.: +81-748-46-2500

**Abstract:** To investigate the effect of paddy-upland (PU) rotation system on greenhouse gas emissions, methane ($CH_4$) and nitrous oxide ($N_2O$) emissions were monitored for three years for a PU rotation field (four cultivations (wheat-soybean-rice-rice) over three years) and continuous paddy (CP) field on alluvial soil in western Japan. Soil carbon storage was also calculated using an improved Rothamsted Carbon (RothC) model. The net greenhouse gas balance was finally evaluated as the sum of $CO_2$eq of the $CH_4$, $N_2O$ and changes in soil carbon storage. The average $CH_4$ emissions were significantly lower and the average $N_2O$ emissions were significantly higher in the PU field than those in the CP field ($p < 0.01$). On $CO_2$ equivalent basis, $CH_4$ emissions were much higher than $N_2O$ emission. In total, the average $CO_2$eq emissions of $CH_4$ plus $N_2O$ in the PU field (1.81 Mg $CO_2$ ha$^{-1}$ year$^{-1}$) were significantly lower than those in the CP field (7.42 Mg $CO_2$ ha$^{-1}$ year$^{-1}$) ($p < 0.01$). The RothC model revealed that the changes in soil carbon storage corresponded to $CO_2$eq emissions of 0.57 and 0.09 Mg $CO_2$ ha$^{-1}$ year$^{-1}$ in the both fields, respectively. Consequently, the net greenhouse gas balance in the PU and CP fields were estimated to be 2.38 and 7.51 Mg $CO_2$ ha$^{-1}$ year$^{-1}$, respectively, suggesting a 68% reduction in the PU system. In conclusion, PU rotation system can be regarded as one type of the climate-smart soil management.

**Keywords:** greenhouse gases emissions; methane; nitrous oxide; paddy-upland rotation; soil carbon storage

## 1. Introduction

In recent years, the average temperature has risen worldwide and global warming has been pointed out, and it is necessary to drastically and sustainably reduce greenhouse gases (GHG) emissions in order to curb climate change [1]. Agriculture is regarded as a GHG emission sector combined with forestry and other land use (AFOLU sector), and their emissions are estimated to account for about one-quarter of total anthropogenic emissions [1]. It has been pointed out that the greenhouse gases $CH_4$ and $N_2O$ emitted from agricultural lands have increased drastically over the past half century due to the expansion of paddy field cultivation worldwide and the increase in nitrogen fertilizer application [2,3]. Since about 90% of paddy rice is produced in Asia [4], various studies have been conducted in Asia to reduce $CH_4$ emissions from paddy fields [5–10]. As one of the techniques for reducing $CH_4$ emissions from paddy fields, effectiveness of water

management such as intermittent irrigation and extension of the mid-summer drainage period have been clarified [11–13]. In this context, rice (*Oryza sativa*) cultivation systems with extended non-flooded period are expected to be effective for reducing $CH_4$ emissions.

A paddy-upland (PU) rotation system is a land-use system that rotates rice, wheat, and soybean within several years to adjust rice production and increase the production of main crops such as wheat and soybean. Conversion from paddy to upland has several advantages, such as improved water permeability by upland farming, the prevention of excessive soil reduction, and increased soil nitrogen availability. It is also known that conversion from upland to paddy can alleviate the reduction of organic matter and related nutrients, acidification of soil, and proliferation of weeds [14].

Kumagai and Konno [15], Nishimura et al. [16], Nishimura et al. [17], and Shiono et al. [18] reported the effect of PU rotation system on the reduction of methane ($CH_4$) emissions. In these reports, $CH_4$ emissions were remarkably reduced especially in the first year of restored (re-converted) paddy fields by PU rotation systems compared to that in continuous rice (CP) paddy fields; this reduction effect was maintained into the second year of re-converted paddy fields. By contrast, it was reported that dinitrogen monoxide (nitrous oxide: $N_2O$) emissions were increased by upland conversion [17–19]. However, reports on the emissions of both $CH_4$ and $N_2O$ over several years under PU rotation system are still limited [17,18]. In addition, PU rotation system is known to affect the soil carbon storage. For example, Sumida et al. [20] reported that a longer PU rotation induced a greater degradation of soil organic matter. Shirato et al. [21] reported that PU rotation enhanced $CO_2$ emissions compared to those in CP cropping. Nevertheless, very few studies have reported the net GHG emissions of PU systems in comparison to CP cropping systems based on direct measurement of both $CH_4$ and $N_2O$ emissions and changes in soil carbon storage. From the viewpoint of GHG balance between GHG emissions and carbon storage at the PU field, Takakai et al. [22] reported the effects of compost application on the net GHG balance from the PU field in relatively cold north-eastern regions of Japan (PU systems: three years for soybean and three years for rice in lysimeter paddy fields). In this study, net GHG balance was evaluated by integrating $CH_4$ and $N_2O$ emissions and carbon dioxide ($CO_2$) emissions calculated from a decline in soil carbon storage. It has been reported that the major component of net GHG emission was $CO_2$ (82–94%) during the soybean cultivation and $CH_4$ (72–84%) during the rice cultivation. Additionally, net GHG emissions during the soybean and rice cultivations were comparable. It is therefore very meaningful and important to quantitatively evaluate the net GHG emissions under PU rotation system on a range of climatic conditions. The objectives of this research were, therefore, to evaluate the effect of PU rotation system on (1) GHG ($CH_4$ and $N_2O$) emissions, (2) changes in soil carbon storage, and (3) the net GHG balance ($CO_2$ equivalents) in comparison with CP system, in relatively warm western Japan.

## 2. Materials and Methods

### 2.1. Experimental Plots and Design

The experiment was conducted in two adjacent fields: a CP field (area: 850 $m^2$) and a PU rotation field (area: 1350 $m^2$) in the Shiga Agricultural Technology Promotion Center (Shiga Agricultural Technology Center) in Omi-Hachiman City, Shiga Prefecture, western part of Japan (35°18′ N, 136°1′ E). In the CP field, rice has been cultivated once a year since 1975. In the PU field, four crop cultivations over three years (i.e., wheat-soybean-rice-rice) have been carried out since wheat cultivation started after the rice harvest in 2003. This PU rotation system has been established as the main cultivation system in Shiga Prefecture.

As shown in Table 1, a comparative experiment of the two cropping systems was carried out for a single three-year rotation period from October 2012 to October 2015. In the PU field, wheat was grown from late October after the harvest of rice in 2012, and soybean was grown from late June 2013, after the wheat cropping. Then, the paddy field was restored in 2014, and rice was cultivated in 2014 and 2015. In the CP fields, rice was cultivated in 2013, 2014 and 2015. In both the PU and CP fields, the underdrains were

installed at 70 cm depth with intervals of 7 to 10 m. In the PU field, the mole drains were also installed perpendicular to the underdrains before wheat cultivation at about 30 cm depth, with intervals of 3 to 5 m. The soil of the experimental fields was classified as Gleyic Fluvisols according to the World Reference Base Classification [23], or as fine-textured Gray Lowland soil according to the Classification of Cultivated Soil in Japan, Third Approximation [24]. Table 2 shows the selected soil chemistry properties before the experiment (2012). In PU field, the values of T-C and T-N were lower than the values in CP, which was considered due to the influence of three rounds of PU rotation until the start of the survey. Detail information on the estimation of temporal change in T-C using Roth C model is described in Section 2.4. The soil pH, Available N, CEC, and $K_2O$ were within range of the improvement target value of Shiga Prefecture in all three years.

**Table 1.** Setting of the experimental plots.

| Treatment [†] | 2012~2013 (First Year) | | 2013~2014 (Second Year) | | 2014~2015 (Third Year) | |
|---|---|---|---|---|---|---|
| CP (continuous Paddy) | Fallow period of paddy (9 October 2012– 13 May 2013) | Paddy rice (13 May 2013– 12 September 2013) | Fallow period of paddy (12 September 2013– 22 April 2014) | Paddy rice (22 April 2014– 17 September 2014) | Fallow period of paddy (17 September 2014– 7 May 2015) | Paddy rice (7 May 2015– 27 October 2015) |
| PU (paddy-upland rotation) | Wheat (9 October 2012– 1 July 2013) | Soybean (1 July 2013– 18 Novermber 2013) | Post soybean fallow period (18 Novermber 2013– 22 April 2014) | Paddy rice (22 April 2014– 17 September 2014) | Fallow period of paddy (17 September 2014– 7 May 2015) | Paddy rice (7 May 2015– 27 October 2015) |

[†] Previous crop (2012): Paddy rice.

**Table 2.** Physicochemical properties of the soils in the experimental plots [1].

| Treat-ment | pH (H$_2$O) | T-C (g kg$^{-1}$) | T-N (g kg$^{-1}$) | Available N [2] (mg kg$^{-1}$) | Available P$_2$O$_5$ (mg kg$^{-1}$) | Available SiO$_2$ (mg kg$^{-1}$) | Free Fe$_2$O$_3$ (g kg$^{-1}$) | CEC (cmol$_c$ kg$^{-1}$) | Exchangeable Base | | | Three Phases Distribution [3] | | | Bulk Density [3] (g cm$^{-3}$) |
|---|---|---|---|---|---|---|---|---|---|---|---|---|---|---|---|
| | | | | | | | | | Ca | Mg | K | Gaseous Phase | Liquid Phase | Solid Phase | |
| | | | | | | | | | (cmol$_c$ kg$^{-1}$) | | | (%) | | | |
| CP | 5.4 | 25.0 | 2.26 | 36.3 | 55.0 | 122.7 | 27.6 | 21.6 | 9.60 | 2.25 | 0.43 | 14.3 | 52.7 | 33.0 | 1.11 |
| PU | 5.9 | 22.6 | 1.95 | 31.5 | 95.0 | 165.7 | 24.5 | 21.4 | 11.29 | 3.66 | 0.56 | 24.1 | 40.0 | 36.0 | 1.01 |

[1] In Oct 2012, five soil samples were collected from the surveyed field, air-dried, adjusted with a 2.0 mm sieve, and subjected to soil chemical analysis. [2] Soil was collected before the rice cultivation in 2015. [3] In Nov 2013 (after soybean harvest), the soil from both treatments was collected at three locations with 100 mL cores, subjected to analysis, and shown as an average value.

In this experiment, two treatments, CP and PU, were established. Three sub-plots were prepared for each treatment, which were located at the central part in each field. The area of each sub-plot was 33.3 m$^2$ for the CP field and 46.0 m$^2$ for the PU field. The size of sub-plot was different between CP and PU, reflecting the planting style, because wheat and soybean were cultivated on ridges, which had longer width than the width of the rows of rice. In the PU field, both wheat and soybean were grown based on the Cultivation Technology Guideline of Shiga Prefecture [25]. The wheat cultivar was 'Nourin No. 61'. The soybean cultivar was 'Kotoyutaka'. The method used was narrow line non-intertillage cultivation (narrow ridge dense planting cultivation method). In both CP and PU fields, rice was cultivated based on the environmentally friendly crop cultivation standard of Shiga Prefecture [26]. The rice cultivar cultivated was 'Mizukagami'. N fertilizer with 50% organic N was used. Water management was 3–5 cm shallow water management from puddling to mid-summer drainage, followed by mid-summer drainage for 7–12 days from mid-June to June, intermittent irrigation from mid-summer drainage to pre-harvest drainage, and water drainage in late August. Rice was harvested in early September. Details of cropping management for paddy rice and wheat and soybean are shown in Tables S1 and S2, respectively.

### 2.2. Measurement of CH$_4$ and N$_2$O Emissions

The CH$_4$ and N$_2$O gas fluxes between the ground surface and the atmosphere in each sub-plot were measured by the closed-chamber method [27]. Gas was basically collected once a week between 9:00 a.m. and 12:00 a.m. Specially, the frequency of measurement

was about two to three times per week for the puddling and transplanting stages, before and after mid-summer and pre-harvest drainages in the paddy rice cultivation period, and immediately after the fertilization of wheat and soybean. In contrast, the measurement frequency was biweekly only from January to March in the post-soybean fallow period and the fallow period of paddy.

An acrylic chamber (60 cm long × 30 cm wide × 50 cm high) was used for the measurement of each sub plot. Inside the chamber, a fan for air agitation and a Tedlar bag for pressure control were installed. The chamber base was inserted into the soil at up to 8 cm depth, to cover two lines of wheat, two strains of soybean, and three strains of rice. The height of the chamber was changed to 100 cm with the growth of the crop, i.e., from early May before topdressing at the ripening stage to harvest for wheat, from mid-August to harvest for soybean, and from early July before topdressing at the panicle formation stage to harvest for rice. For gas sampling, the gas in the chamber was mixed several times with a 50 mL syringe, and then 30 mL was sampled three times at 10-min intervals. The gas concentration was analyzed based on the method of Sudo [28] using a gas chromatograph (GC; GC-14A, Shimadzu, Kyoto, Japan) with a flame ionization detector and electron capture detector. The cumulative $CH_4$ and $N_2O$ emissions were calculated using the trapezoidal integration method. Cumulative emissions of each gas were converted to cumulative $CO_2$eq emissions ($CH_4$ and $N_2O$ combined) based on the global warming potential (GWP) conversion factor of 28 for $CH_4$ and 265 for $N_2O$ [29].

The air temperature inside the chamber and the soil temperature (5 cm deep) at the time of collection were measured with a temperature data logger (Ondotori, T & D, Matsumoto, Japan).

### 2.3. Measurement of Selected Field Properties

Selected soil properties were investigated based on the Soil Nutrient Measurement Method Committee [30] and the Soil Environment Analysis Method Editing Committee [31]. Free iron oxide was extracted by acetate buffer extraction at pH 2.8 [32] and quantified with an atomic absorption spectrophotometer. The available phosphate was measured using the Truog method [33], and the available silica was measured by acetate buffer extraction at pH 4. For available nitrogen, soil samples collected from the plow layer just before puddling in 2015 were incubated for four weeks at 30 °C, and the amount of ammonium ion was determined using the indophenol method.

In the periods other than rice cultivation, the volumetric moisture content (0–12 cm deep) was measured at five locations around the chamber using a portable soil moisture meter (HydroSense, Campbell, UT, USA). From the volumetric moisture content and solid phase ratio (33.0–36.4%), the water-filled pore space (WFPS), defined as the ratio of water volume to the soil void volume, was calculated, and the average value for each plot was determined. For cases in which the calculated values exceeded 100%, they were treated as 100% under saturation. For the cumulative precipitation, we used meteorological observation data from the Shiga Agricultural Technology Center. The redox potential ($E_h$) was measured at the time of gas sampling of the rice cultivation. The measurement was carried out by connecting a portable $E_h$ meter (PRN-41; Fujiwara Seisakusho, Tokyo, Japan) to a platinum electrode embedded at a depth of 5 cm; measurements were made at four locations around the chamber, and the average value was determined.

### 2.4. Estimation of Soil C Storage Based on the RothC Model

The change in soil carbon is slow and there may be large spatial variations, and it takes a long time to detect the change by actual measurement. In this study, changes in soil carbon content were calculated using the RothC model, which had been verified to be simple and accurate [34–36].

The temporal changes in the soil carbon contents of the CP and PU fields were predicted, using the past data of the survey fields, with the improved RothC model for paddy field soil [34–36]. The model calculation was carried out for the data starting in 1995, when

the soil C concentration data for the survey fields was available (25.2 g kg$^{-1}$). The initial soil carbon stock in 1995 was calculated to be 37.5 tC ha$^{-1}$ based on the plow layer depth and the bulk density of 15 cm and 1.0 g cm$^{-3}$, respectively. The clay content in the plow soil layer was set as 31.3% [37]. For the precipitation and temperature, we used meteorological observation data (1995–2015) from the Shiga Agricultural Technology Center. The amount of evapotranspiration was estimated using the method of Thornthwaite (1948) [38] with the temperature data, and converted to open pan evaporation, which is required for the model calculation, by dividing it by 0.75. Data including the C concentration; bulk density of the plow soil layer; crop yields of paddy rice, wheat, and soybean were collected from the research results summary for Shiga Prefecture and the Special Research Report of the Shiga Agricultural Research Center Agricultural Experiment Station [39]. Missing values were replaced by the mean of the preceding and proceeding values. The amount of carbon input from crop residues, such as the stubble and roots, was converted from the crop yield data, using the data on the dry matter weight ratio of various crops [34,40]. As the annual rate of soil carbon stock, average values of the above mentioned 12 years from 1995 to 2017 were calculated for both PU and CP fields. This is because long-term data are often more favorable to understand and analyze the relationship between farmland management and the increase or decrease in soil carbon [41] due to relatively small temporal changes and relatively large spatial variation. RMSE (The root mean square error) were calculated from the differences between predicted values by the RothC model and the measured values for each field. The RMSEs in the PU and CP fields were 2.64 and 2.39, respectively, suggesting similar and relatively small error in both fields.

### 2.5. Estimation of the Net GHG Balance

Finally, the net GHG balance was calculated as the sum of GHG emissions and soil carbon storage during the study period. Annual CO$_2$eq data of both CH$_4$ and N$_2$O emissions were used as the data of GHG emissions, and the amount of soil carbon stored annually in the CP and PU fields were regarded as soil carbon storage and converted to the CO$_2$eq emissions. The calculation formula is as follows.

Net GHG balance (Mg CO$_2$eq ha$^{-1}$ year$^{-1}$) = GHG emissions + CO$_2$eq emissions

If the value was positive, it suggested a net emission, and if the value was negative, it suggested a net absorption. The mitigation effect was evaluated by subtracting the CP value from the PU value.

### 2.6. Statistical Analysis

Two-way analysis of variance with treatment and year as factors (ANOVA) was performed for the annual CH$_4$, N$_2$O and their CO$_2$eq emissions. In this study, differences with $p < 0.05$ were considered significant [42].

## 3. Results

### 3.1. CH$_4$, N$_2$O, and Their Carbon Dioxide Equivalent (CO$_2$eq) Emissions

The CH$_4$ and N$_2$O flux in the three years are shown in Figure 1a,b, respectively, and seasonal changes in precipitation and water-filled pore space (WFPS), and E$_h$ in the three years of the experiment are shown in Figure 1c,d, respectively. CH$_4$, N$_2$O, and CO$_2$eq emissions for the three years are summarized in Table 3.

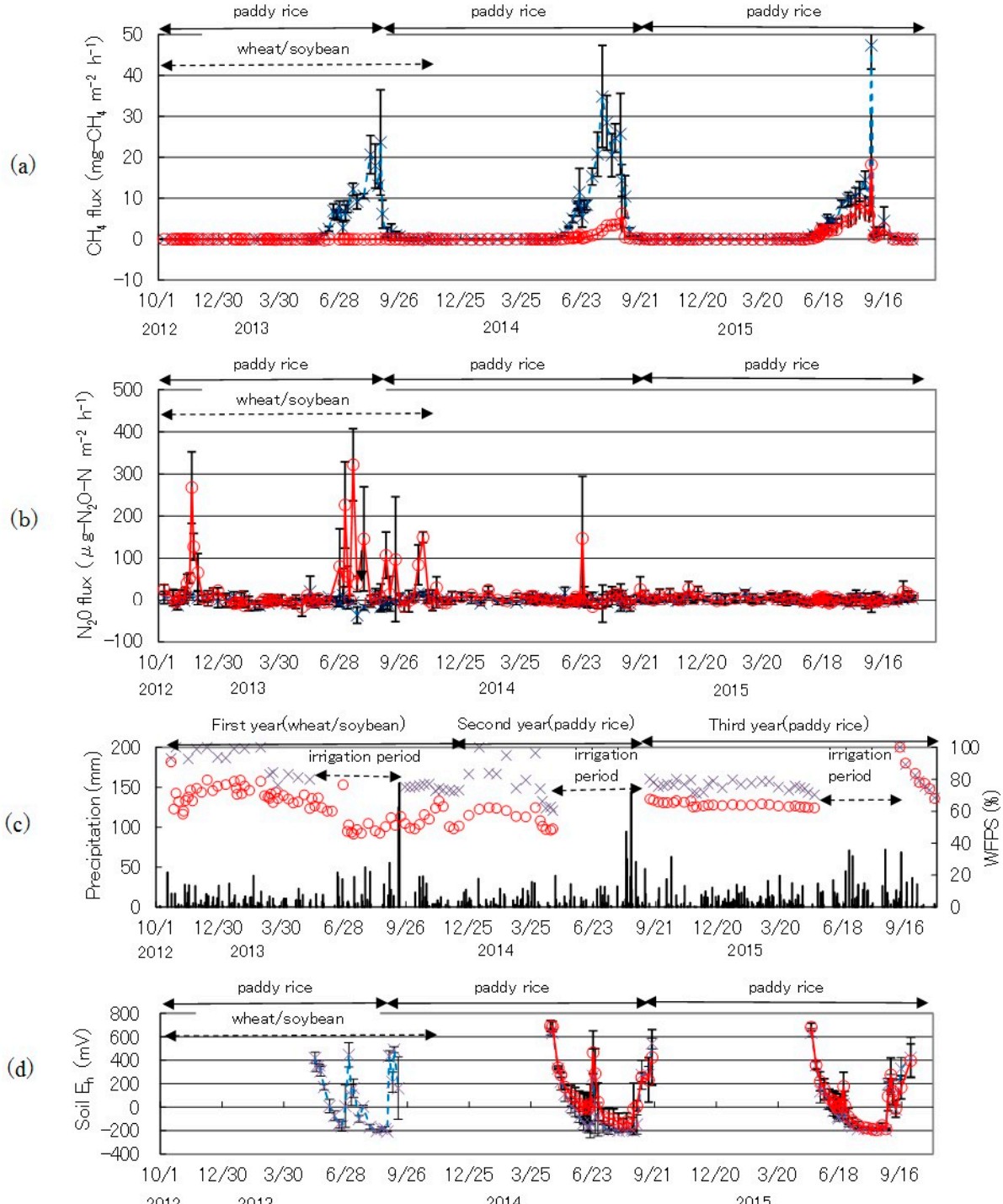

**Figure 1.** Changes in (**a**) CH$_4$ flux, (**b**) N$_2$O flux, (**c**) Precipitation and WFPS and (**d**) Soil E$_h$, in the three years. ×(Blue): CP, ○(Red): PU, (c) ┃: precipitation. Error bar is shown as standard deviation. The data in PU was derived from the previous report (Hasukawa et al. [43]).

**Table 3.** Annual $CH_4$, $N_2O$ and their $CO_2$eq emissions during the survey period (2012–2015).

| Year | Treatment | $CH_4$ Emission (kgCH4-C ha$^{-1}$ year$^{-1}$) | | $N_2O$ Emission (kgN2O-N ha$^{-1}$ year$^{-1}$) | | GWP ($CO_2$eq) [†] | | | | | |
|---|---|---|---|---|---|---|---|---|---|---|---|
| | | | | | | $CH_4$ (Mg ha$^{-1}$ year$^{-1}$) | | $N_2O$ (Mg ha$^{-1}$ year$^{-1}$) | | Total (Mg ha$^{-1}$ year$^{-1}$) | |
| 2012–2013 (First year) | CP | 165.3 ± 14.8 | A | −0.169 ± 0.185 | A | 6.17 ± 0.55 | A | −0.07 ± 0.08 | A | 6.10 ± 0.48 | A |
| | PU | −0.6 ± 1.2 | B | 2.591 ± 0.322 | B | −0.02 ± 0.05 | B | 1.08 ± 0.13 | B | 1.05 ± 0.18 | B |
| 2013–2014 (Second year) | CP | 273.5 ± 28.4 | A | 0.113 ± 0.085 | A | 10.21 ± 1.06 | A | 0.05 ± 0.04 | A | 10.26 ± 1.03 | A |
| | PU | 28.4 ± 5.3 | B | 0.340 ± 0.338 | A | 1.06 ± 0.20 | B | 0.14 ± 0.14 | A | 1.20 ± 0.30 | B |
| 2014–2015 (Third year) | CP | 155.4 ± 7.9 | A | 0.263 ± 0.109 | A | 5.80 ± 0.29 | A | 0.11 ± 0.05 | A | 5.91 ± 0.33 | A |
| | PU | 81.1 ± 31.1 | B | 0.342 ± 0.259 | A | 3.03 ± 1.16 | B | 0.14 ± 0.11 | A | 3.17 ± 1.17 | B |
| 2012–2015 (Average) | CP | 198.1 ± 11.9 | | 0.069 ± 0.109 | | 7.39 ± 0.44 | | 0.03 ± 0.05 | | 7.42 ± 0.40 | |
| | PU | 36.3 ± 11.1 | | 1.091 ± 0.219 | | 1.36 ± 0.42 | | 0.45 ± 0.09 | | 1.81 ± 0.38 | |
| Analysis of variance | Treatment | ** | | ** | | ** | | ** | | ** | |
| | Year | ** | | ** | | ** | | ** | | ** | |
| | Interaction | ** | | ** | | ** | | ** | | ** | |

The values are shown as mean ± standard deviation. [†] The equation for calculating combined GWP is described in the text. The analysis of variance was performed in the treatment and year two-way allocation (**: significant difference at the 1% level). Numbers within a column followed by different letters differ significantly among the treatment (*t*-test, $p < 0.05$).

### 3.1.1. $CH_4$ and $N_2O$ Flux

The $CH_4$ flux in the rice cultivation increased to the mid-summer drainage in both treatments, and temporarily decreased during the mid-summer drainage. In the CP field, the maximum values at the time of mid-summer drainage were 7.9, 11.5, and 4.6 mg $CH_4$ m$^{-2}$ h$^{-1}$ in the first, second and third year, respectively. In the PU field, they were 1.0 and 2.3 mg $CH_4$ m$^{-2}$ h$^{-1}$ in the second and third year, respectively, which was much lower than those in the CP field. It increased again with watering after the mid-summer drainage, and then decreased immediately after pre-harvest drainage. In the CP field, the maximum values were 23.6, 34.9, and 47.3 mg $CH_4$ m$^{-2}$ h$^{-1}$ in the first, second and third year, respectively. In the PU field, they were 6.2, and 18.2 mg $CH_4$ m$^{-2}$ h$^{-1}$ in the second and third year, respectively, which were again much lower than those in the CP field, in the third. The $CH_4$ flux after the mid-summer drainage in both treatments tended to be higher than before the mid-summer drainage. In total, the $CH_4$ flux in the CP field remained at a high level throughout the experimental period compared to that in the PU field. In the PU field, the $CH_4$ flux tended to be lower in the first year than in the second year of paddy rice cultivation after upland crops. The $CH_4$ flux in the wheat and soybean cultivations was kept at around 0 mg m$^{-2}$ h$^{-1}$ throughout the experimental period in the PU field.

The $N_2O$ flux increased significantly just after the base fertilization in both wheat and soybean cultivation in the PU field. The maximum values were 267.5 μg N m$^{-2}$ h$^{-1}$ for wheat cultivation and 321.9 μg N m$^{-2}$ h$^{-1}$ for soybean cultivation. In the soybean cultivation, the $N_2O$ flux after rainfall increased several times. In PU, the maximum value at the time of mid-summer drainage was 147.2 μg N m$^{-2}$ h$^{-1}$. It did not increase significantly during the additional fertilization of wheat, topdressing at the panicle formation stage, and topdressing at ripening stage. In the other period, it was maintained at around 0 μg N m$^{-2}$ h$^{-1}$. In the CP field, the $N_2O$ flux was maintained at around 0 μg N m$^{-2}$ h$^{-1}$ throughout the three years.

### 3.1.2. Precipitation

The cumulative precipitation in 2013, 2014 and 2015 was 1520, 1583 and 1800 mm, respectively. The mean air temperature in 2013, 2014 and 2015 the first year was 14.7, 14.4 and 15.0 °C, respectively. The climate was relatively warm with little annual difference in both precipitation and temperature.

### 3.1.3. Water-Filled Pore Space

The WFPS at the time of gas collection in the first year ranged from 46 to 91% (average 64%) in the PU field. In the CP field, the WFPS remained at 79 to 100% (average 92%) during the non-flooding period before the inflow of paddy rice cultivation. The WFPS during the non-flooding period until the second year of the inflow of paddy rice cultivation was 48 to 62% (average 56%) in the PU field and 60 to 100% (average 78%) in the CP field.

The WFPS during the non-flooding period until the third year of inflow of paddy rice cultivation was 62 to 67% (average 64%) in the PU field and 70 to 80% (average 76%) in the CP field. The WFPS from the harvest of rice to the end of the survey remained at 68 to 100% (82% on average) in the PU field, and 68 to 100% (81% on average) in the CP field. It was high just after harvesting rice in both treatments but tended to decrease gradually. In all three years, the WFPS in both treatments increased in winter when the precipitation and temperature decreased.

### 3.1.4. Soil $E_h$

In the CP field, the soil $E_h$ decreased rapidly with the start of flooding and continued to decrease until the start of mid-summer drainage in all three years. It showed a positive value with the mid-summer drainage and became oxidative, but it decreased gradually with the incoming water after the mid-summer drainage and remained at around $-200$ mV until pre-harvest drainage. After rice harvesting, it showed a positive value and remained at around 600 mV. In the PU field, the soil $E_h$ decreased gradually with the start of spring in the PU field and remained at around 0 mV before the mid-summer drainage in the second and third years.

### 3.1.5. Annual $CH_4$, $N_2O$, and Their $CO_2$eq Emissions

The annual $CH_4$ emissions in the CP field were $165.3 \pm 14.8$, $273.5 \pm 28.4$, and $155.4 \pm 7.9$ kg $CH_4$-C ha$^{-1}$ year$^{-1}$ in the first, second and third year, respectively (Table 3). In the PU field, the annual $CH_4$ emissions were $-0.6 \pm 1.2$, $28.4 \pm 5.3$, and $81.1 \pm 31.1$ kg $CH_4$-C ha$^{-1}$ year$^{-1}$ in the first, second (first year of paddy rice cultivation after upland crops) and third year (second year of paddy rice cultivation after upland crops), respectively. The annual $CH_4$ emissions in the PU field were significantly lower than those in the CP field in each of the three years (in the first and second year: $p < 0.01$, in the third year: $p < 0.05$). On average, therefore, annual $CH_4$ emissions in the PU field was significantly lower than those in the CP field ($p < 0.01$).

The annual $N_2O$ emissions in the CP field were $-0.169 \pm 0.185$, $0.113 \pm 0.085$, and $0.263 \pm 0.109$ kg $N_2O$-N ha$^{-1}$ year$^{-1}$ in the first, second and third year, respectively. In the PU field, the annual $N_2O$ emissions were $2.591 \pm 0.322$, $0.340 \pm 0.338$, and $0.342 \pm 0.259$ kg $N_2O$-N ha$^{-1}$ year$^{-1}$ in the first, second and third year, respectively. The annual $N_2O$ emissions in the PU field were significantly higher than those in the CP field in the first year ($p < 0.01$) and were not significantly different in the second and third years. On average, therefore, annual $N_2O$ emissions in the PU field was significantly higher than those in the CP field ($p < 0.01$).

The annual $CO_2$eq-converted $CH_4$ emissions in the CP field were $6.17 \pm 0.55$ Mg, $10.21 \pm 1.06$ Mg, and $5.80 \pm 0.29$ Mg ha$^{-1}$ year$^{-1}$ in the first, second and third year, respectively (Table 3). In the PU field, the annual $CO_2$eq-converted $CH_4$ emissions were $-0.02 \pm 0.05$ Mg, $1.06 \pm 0.20$ Mg, and $3.03 \pm 1.16$ Mg ha$^{-1}$ year$^{-1}$ in the first, second and third year, respectively. The annual $CO_2$eq-converted $CH_4$ emissions in the PU field were significantly lower than those in the CP field in each of the three years (in the first and second year: $p < 0.01$, in the third year: $p < 0.05$). On average, therefore, annual GWP of $CH_4$ emissions in the PU field was significantly lower than those in the CP field ($p < 0.01$).

The annual $CO_2$eq-converted $N_2O$ emissions in the CP field were $-0.07 \pm 0.08$ Mg, $0.05 \pm 0.04$ Mg, and $0.11 \pm 0.05$ Mg ha$^{-1}$ year$^{-1}$ in the first, second and third year, respectively (Table 3). In the PU field, the annual $CO_2$eq-converted $N_2O$ emissions were $1.08 \pm 0.13$ Mg, $0.14 \pm 0.14$ Mg, and $0.14 \pm 0.11$ Mg ha$^{-1}$ year$^{-1}$ in the first, second and third year, respectively. The annual $CO_2$eq-converted $N_2O$ emissions in the PU field were significantly higher than those in the CP field only in the first year ($p < 0.01$). On average, therefore, annual $CO_2$eq-converted $N_2O$ emissions in the PU field was significantly higher than those in the CP field ($p < 0.01$).

Based on the $CO_2$eq data of both $CH_4$ and $N_2O$, the annual overall $CO_2$eq emissions in the CP field were calculated as $6.10 \pm 0.48$, $10.26 \pm 1.03$, and $5.91 \pm 0.33$ Mg ha$^{-1}$

year$^{-1}$ in the first, second and third year, respectively. In the PU field, the annual overall $CO_2$eq emissions were $1.05 \pm 0.18$, $1.20 \pm 0.30$, and $3.17 \pm 1.17$ Mg ha$^{-1}$ year$^{-1}$ in the first, second and third year, respectively. The annual overall $CO_2$eq emissions in the PU field were significantly lower than those in the CP field in each of the three years (in the first and second year: $p < 0.01$, in the third year: $p < 0.05$). On average, therefore, the annual overall $CO_2$eq emissions in the PU field was significantly lower than those in the CP ($p < 0.01$). The relative percentage of annual $CH_4$ emissions to annual overall $CO_2$eq emissions throughout three years was 99.7% in the CP field and 74.9 % in the PU field, respectively.

### 3.2. Predictive Evaluation of Soil Carbon Storage Using the RothC Model

The predicted value of soil carbon storage in the CP field since the start of PU rotation was almost constant, i.e., 0.411 Mg C ha$^{-1}$ in 2003 and 0.408 Mg C ha$^{-1}$ in 2015, whereas, in the PU field, it showed a decreasing trend, i.e., 0.411 Mg C ha$^{-1}$ in 2003 and 0.392 Mg C ha$^{-1}$ in 2015 (Figure 2). The average changes in the amount of soil carbon per rotation were $-0.03$ Mg C ha$^{-1}$ years$^{-1}$ in the CP field, and $-0.15$ Mg C ha$^{-1}$ years$^{-1}$ in the PU field. Accordingly, the slight decreases in the amount of soil carbon in the CP and PU fields corresponded to increases in $CO_2$eq emissions of 0.09 and 0.57 Mg $CO_2$ ha$^{-1}$ year$^{-1}$, respectively.

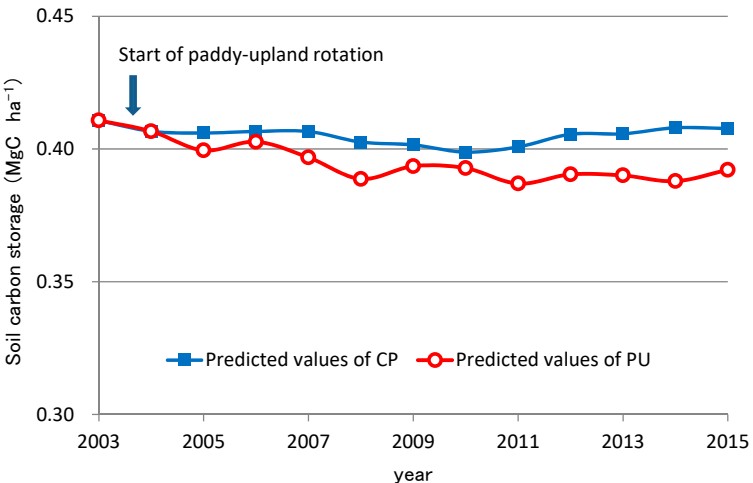

**Figure 2.** Estimation of the changes in the amount of soil carbon by RothC model.

### 3.3. Net GHG Balance

Figure 3 shows the net GHG balance as the sum of both $CH_4$ and $N_2O$ emissions and soil carbon storage. The annual overall GHG emissions were $7.42 \pm 0.40$ Mg $CO_2$eq ha$^{-1}$ year$^{-1}$ in the CP field, and $1.81 \pm 0.38$ Mg $CO_2$eq ha$^{-1}$ year$^{-1}$ in the PU field (Table 3). Slight decrease in the amount of soil carbon corresponded to the emission of 0.09 Mg $CO_2$eq ha$^{-1}$ year$^{-1}$ in the CP field and 0.57 Mg $CO_2$eq ha$^{-1}$ year$^{-1}$ in the PU field, respectively. Accordingly, the net GHG balance was calculated to be 7.51g and 2.38 Mg $CO_2$eq ha$^{-1}$ year$^{-1}$ in the CP and PU field, respectively. As a result, the mitigation effect of net GHG emission by the conversion of land use from CP to PU was evaluated as 5.13 Mg $CO_2$eq ha$^{-1}$ year$^{-1}$, i.e., a 68% reduction. This result strongly supported that the paddy-upland rotation system can be regarded as one type of the climate-smart soil management under the concept of "climate-smart soils" [44].

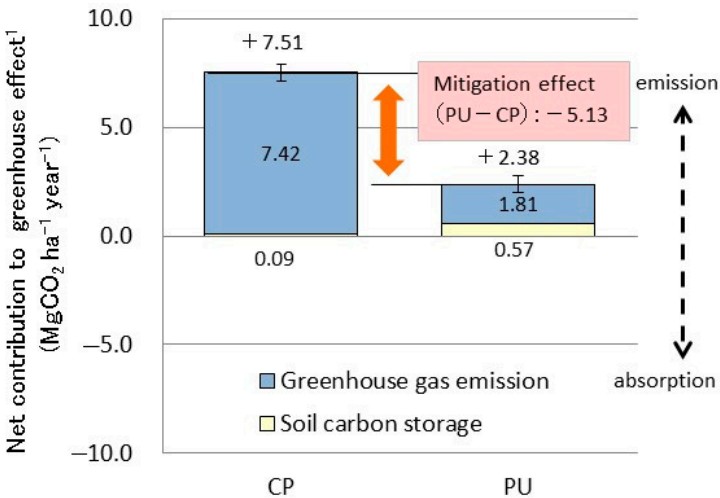

**Figure 3.** GHG emission and absorption in the experimental field. [1] $CO_2$ conversion. Error bar is shown as standard deviation in the GHG emission.

### 3.4. Yield, Quality, and Nitrogen Uptake of Rice, Wheat and Soybean

The yield, quality, and nitrogen uptake of paddy rice, wheat, and soybean are shown in Table 4. The yield of paddy rice was 692 g m$^{-2}$ in 2013, 532 g m$^{-2}$ in 2014, and 580 g m$^{-2}$ in 2015 in the CR field. In the PU field, it was 552 g m$^{-2}$ in 2014 and 603 g m$^{-2}$ in 2015. There were no clear differences in both treatments in the yield of paddy rice both in 2014 and 2015. There were no evident differences in appearance quality and brown rice protein content both in 2014 and 2015, nor in the nitrogen uptake of the above-ground parts (unhulled rice + straw). The yield of wheat in the PU field was 507 g m$^{-2}$, and the yield of soybean was 505 g m$^{-2}$, the standard yield [25] was secured. The nitrogen uptake or accumulation of the above-ground parts (wheat: grain + straw, soybean: grain + stem + pod) was 10.2 g N m$^{-2}$ for wheat and 33.8 g N m$^{-2}$ for soybean.

**Table 4.** Yield, quality and nitrogen uptake of paddy rice, wheat and soybean.

| Year | Treatment | Paddy Rice | | | | | | | Wheat Soybean | | | | | | | | | |
| | | Yield | | | Protein Content in Brown Rice [3] | Nitrogen Uptake | | | Yield (Wheat) | | Nitrogen Uptake (Wheat) | | | Yield (Soybean) | | Nitrogen Accumulation (Soybean) | | |
| | | Unhulled Rice Weight [1] | Straw Weight [1] | Refined Brown Rice Weight [2] | | Unhulled rice | Straw | Total | Grain Weight [4] | Straw Weight | Grain | Straw | Total | Grain Weight [5] | Stem and Pod Weight | Grain | Stem and Pod | Total |
| | | (g m$^{-2}$) | | | (%) | (gN m$^{-2}$) | | | (g m$^{-2}$) | | (gN m$^{-2}$) | | | (g m$^{-2}$) | | (gN m$^{-2}$) | | |
| 2012–2013 | CP | 894 | 786 | 692 | 6.2 | 8.2 | 3.8 | 12.0 | — | — | — | — | — | — | — | — | — | — |
| | PU | — | — | — | — | — | — | — | 507 | 444 | 9.0 | 1.2 | 10.2 | 505 | 455 | 30.9 | 2.9 | 33.8 |
| 2013–2014 | CP | 787 | 689 | 532 | 8.0 | 8.7 | 3.9 | 12.6 | — | — | — | — | — | — | — | — | — | — |
| | PU | 820 | 777 | 552 | 8.3 | 9.2 | 4.6 | 13.8 | — | — | — | — | — | — | — | — | — | — |
| 2014–2015 | CP | 731 | 780 | 580 | 7.4 | 7.3 | 4.8 | 12.1 | — | — | — | — | — | — | — | — | — | — |
| | PU | 769 | 775 | 603 | 7.3 | 7.7 | 4.2 | 11.9 | — | — | — | — | — | — | — | — | — | — |

[1] Per air-dry matter. [2] >1.8 mm mesh, the value at the moisture content of 14.5%. [3] Measured by a rice barley analyzer, the value at the moisture content of 14.5%. [4] 2.2 mm or more in grain thickness, converted to at the value 12.5% moisture. [5] 5.5 mm or more in grain diameter, converted to the value at 15.0% moisture.

## 4. Discussion

### 4.1. Temporal Changes in the Mitigation Effect of GHG Emissions and Its Regulatory Factors

The reduction of $CO_2$eq emissions showed a clear temporal trend. As described in the results, the annual overall $CO_2$eq emissions in the PU field and CP were 1.05 and 6.10 in the first year, 1.20 and 10.26 in the second year, and 3.17 and 5.91 Mg ha$^{-1}$ year$^{-1}$ in the third year, respectively. Namely, the mitigation effect of the PU system was 89% in the first year of re-converted paddy (the second year), and 47% in the second year of re-converted paddy (the third year), suggesting that the mitigation effect decreases with the period after the re-conversion to paddy. Additionally, proportion of $CH_4$ in $CO_2$eq emissions was much higher than that of $N_2O$ in both fields. A similar trend was observed in the case of a PU system in relatively cold north-eastern region of Japan [18]. Additionally, Takakai et al. [22] reported that the effect of reducing $CH_4$ emissions by PU rotation was observed at least until the second year after the conversion, and that the effect disappears in the third year in north-eastern Japan. The reduction effect was probably controlled by soil redox conditions. This is because $CH_4$ production in CP began immediately after transplanting when the soil $E_h$ decreased rapidly to $-200$ mV as, while little $CH_4$ emission was observed in PU where the soil $E_h$ reached $-200$ mV only limited period during the two-year period.

In addition, redox condition in soil depend on the WFPS (soil moisture) and application of organic matter during unsubmerged period. The WFPS of the PU field in the post-soybean fallow period and in the paddy fallow period remained at relatively low levels compared to those of the CP field. Shiratori et al. [45] reported that the higher soil moisture before water inflow led to higher $CH_4$ emission rate during rice cultivation. Su et al. [46] reported that by planting winter wheat after rice cultivation from the second crop of rice, drainage is improved and $CH_4$ emissions are reduced. In our survey, the WFPS values in the CP field just before the inflow remained higher in both periods than those in the PU field, which led to increased $CH_4$ emissions in the CP field. Additionally, from the viewpoint of decomposition of applied organic matter, Nishimura et al. [17] reported that $CH_4$ emissions were significantly reduced by PU rotation in the first year of the paddy rice cropping after upland crop cultivation, where there was not much residue incorporated from the previous upland conversion crop; whereas there was no clear difference in the second year of paddy rice cropping after upland crop cultivation in the paddy field tested by lysimeter. In our survey, however, a certain reduction effect of $CH_4$ was observed even in the second year of paddy rice cultivation after upland crops, possibly because the decomposition of rice straw incorporated in autumn was promoted by the low level of WFPS and relatively warm temperature in the fallow period of paddy.

### 4.2. Effect of Climate on the Mitigation Effect of PU Rotation Systems

To our knowledge, there are only limited number of reports on long-term GHG emissions ($CO_2$eq emissions: $CH_4$ and $N_2O$ combined GWP) under PU rotation and CP treatments in Asia, as shown in Table 5. Shiono et al. [18] investigated the emissions in a cooler region in Japan, Cha-un et al. [47] investigated the emissions in Thailand with tropical monsoon climate and this study was conducted in a relatively warm region in Japan. The average temperature was in the order of Cha-un et al. (27.3 °C) > this study (14.9 °C) > Shiono et al. (11.7 °C), and the precipitation was in the order of this study (1529 mm) > Shiono et al. (1238 mm) > Cha-un et al. (1043 mm). The soils investigated were all alluvial soil.

From these three survey cases, a remarkable effect of reducing $CO_2$eq emissions in PU was observed regardless of the difference in planting system, temperature and precipitation. The reduction rate was about 75%, i.e., a 75% reduction in Shiono et al. [18] and Cha-un et al. [47], a 76% reduction in our survey. These results clarified that PU rotation is an extremely effective global warming mitigation technology from paddy fields.

**Table 5.** Comparison of previous studies on GHG emission in PU.

| Report Case | Survey Area | Annual Mean Air Temperature [1] | Annual Precipitation [1] | Soil Type | PU-System [2] | Investigation Period | Harvested Residue | $CH_4$-C Emissions [3] in CP | Average $CO_2$eq Emissions [4] | | Reduction Rate [5] of $CO_2$eq Emissions for PU (%) | | |
|---|---|---|---|---|---|---|---|---|---|---|---|---|---|
| | | | | | | | | | CP | PU | Total | Restored Field (year) | |
| | | | | | | | | $(gCH_4\text{-}C\ m^{-2})$ | $(Mg\ CO_2eq\ ha^{-1}\ year^{-1})$ | | | 1st | 2nd |
| Shiono et al. (2014) | Cold region of Japan | 11.7 °C | 1238 mm | alluvial | S-S-R-R | 4 year | removed | 30.5 | 10.15 | 2.54 | 75 | 83(84) | 37(37) |
| Cha-un et al. (2017) | Thailand | 27.3 °C | 1043 mm | alluvial | C-R-C-R | 2 year | incorporated | 50.3 | 16.97 | 4.30 | 75 | 61(63) [6] | - |
| This report | Warm region of Japan | 14.9 °C | 1529 mm | alluvial | W S-R-R | 3 year | incorporated | 19.8 | 7.42 | 1.81 | 76 | 88(89) | 46(47) |

Hasukawa compiled three case data of Shiono et al. [18], Cha-un et al. [47] and this report. [1] Japan Meteorological Agency annual average over 30 years between 1981 and 2010 in representative point of prefecture. Cha-un et al. [47]: quoted from paper. [2] W: Wheat, S: soybean, R: rice, C: corn. Cha-un et al. [47] CP: double cropping of rice, PU: two crops a year. [3] Average value of investigation period. [4] $CH_4$ and $N_2O$ combined GWP. [5] ( ): Reduction of $CH_4$ emissions. [6] Calculated by the average value of two times of paddy rice after corn.

The ratio of $CH_4$ reduction in the first year of re-converted rice was large in all cases, i.e., an 84% reduction in Shiono et al. [18], a 63% reduction in Cha-un et al. [47] and an 89% reduction in our survey, which then decreased to 37% in Shiono et al. [18] and 46% in our survey, respectively, in the second year of re-converted rice. In our survey, soybean residue was incorporated, but a similar magnitude of GHG emission reduction was obtained by PU rotation. This could be due to the fact that the WFPS in the PU field after incorporation was lower than that in the CP field, and the aerobic decomposition of organic matter progressed further in the PU field. In Cha-un et al. [47], the same $CO_2$eq emission reduction effect has been obtained even in the field where the residue was incorporated in two crops a year with a large amount of biomass input. This would be because decomposition of organic matter in Thailand proceeded much faster than the other two cases due to more oxidative state, reflecting higher temperature and lower precipitation.

From the viewpoint of soil carbon storage, Takakai et al. [22] reported that the decrease in soil carbon storage is larger in soybean cultivation than in paddy rice cultivation, which may be mainly due to the difference in redox condition between paddy-upland fields and paddy fields. In our survey, the soil carbon content was maintained in the CP field, but decreased in the PU field based on the RothC model. These facts were also consistent with the report that the PU field has disadvantage in terms of soil carbon storage [20].

Based on these results, the importance to evaluate both the $CO_2$eq emission reduction effect and soil carbon storage was demonstrated to understand overall effect of PU rotation system, or any agricultural system in general, on GHG emissions.

*4.3. Comprehensive Assessment of Soil Carbon Stocks and GHG Emissions*

To find out which of the three GHG components ($CH_4$, $N_2O$, and $CO_2$) has the greatest contribution to the greenhouse effect from farmland, it is necessary to investigate the emissions of all three components in the field. However, there are only a few comprehensive evaluation cases of measuring three GHGs in farmland. In our study, the effect of PU on reducing GHG emissions in a warmer region in Japan was evaluated for the first time from the perspective of net GHG emission balance by integrating $CH_4$ and $N_2O$ emissions and $CO_2$ emissions calculated from a decline in soil carbon storage.

As an example of simultaneous evaluation of three component GHGs in paddy fields, Ishibashi et al. [48] reported that $CH_4$ had the greatest contribution to emissions among the three GHGs based on the results of research on a non-tilled direct seeding cultivation of paddy rice in a warmer region in Japan. Additionally, Takakai et al. [22] reported that three components were evaluated simultaneously in soybean and paddy rice after paddy-upland field rotation in a cooler region in Japan, and it was reported that $CH_4$ emissions were the highest in paddy rice and $CO_2$ emissions were the highest in soybean.

In addition, Takakai et al. [22] reported that a large amount of $CO_2$ emissions are generated in soybean fields, especially when immature compost is added, and that the increase in these $CO_2$ emissions may offset the reduction in $CH_4$ emissions by PU rotation,

so it was considered necessary to require appropriate organic matter management during paddy-upland rotation. Our results also suggested that $CH_4$ had the greatest contribution to emissions among the three GHGs.

This study revealed that the PU rotation treatment reduced the total $CO_2$eq emissions by about 68% because of the reduction of $CH_4$ emissions in the first and second years of re-converted rice, while the PU treatment had no effect on the yield and quality. Our results also suggested that reducing $CH_4$ emissions would be most effective for reducing GHG emissions from paddy fields. To further reduce the contribution to global warming by PU rotation systems, therefore, the introduction of GHG mitigation measures on water management, such as extending the period of mid-summer drainage for $CH_4$ reduction technology [12] should be considered. In fact, GHG emissions (cumulative $CO_2$eq emissions) were reduced significantly by the extension of the mid-summer drainage period by one week in paddy rice cultivation in combination with the use of coated fertilizer in wheat and soybean cultivations [43]. Further reduction of GHG emissions in PU rotation system by introducing these mitigation measures would be desirable to contribute to the mitigation of global warming.

## 5. Conclusions

A three year field experiment was carried out to investigate the overall greenhouse effects in the paddy-upland (PU) rotation, i.e., four cultivations (wheat-soybean-rice-rice) over three years, and in the paddy continuous rice (CP) cropping over three years. Total $CO_2$eq emissions ($CH_4$ and $N_2O$ combined global warming potential [GWP]) in the PU field (1.81 Mg $CO_2$eq ha$^{-1}$ year$^{-1}$), were 76% lower than those in the CP field (7.42 Mg $CO_2$eq ha$^{-1}$ year$^{-1}$). Soil carbon budget calculated using an improved Rothamsted Carbon (RothC) model indicated that decrease in soil carbon in the PU and CP fields corresponded to $CO_2$eq emissions of 0.57 and 0.09 Mg $CO_2$eq ha$^{-1}$ year$^{-1}$, respectively. Accordingly, the overall greenhouse effects in the PU and CP fields were estimated to be 2.38 and 7.51 Mg $CO_2$eq ha$^{-1}$ year$^{-1}$, respectively. Our research revealed, therefore, that a 68% reduction of the overall greenhouse effect was possible by the conversion from the CP field to the PU field. In conclusion, the finding that the PU treatment can reduce overall greenhouse effect considerably compared to the CP treatment would be applicable to paddy rice production systems in a range of areas not only in Japan but also widely in monsoon Asia.

**Supplementary Materials:** The following are available online at https://www.mdpi.com/2077-0472/11/1/52/s1, Table S1: Outline of paddy rice cropping management, Table S2: Outline of wheat and soybean cropping management.

**Author Contributions:** H.H., Y.I. and S.T. performed the experiments; H.H., S.S., N.O., T.S. and Y.S. conceived and designed the experiments; H.H. and Y.I. analyzed the data; H.H. wrote the paper; J.Y. assisted consideration and conclusion on this manuscript; Y.S. and J.Y. gave many constructive comments on this manuscript. All authors have read and agreed to the published version of the manuscript.

**Funding:** This study was supported by a project entitled "Basic Survey on Farmland Soil Greenhouse Gas Emissions" in the Ministry of Agriculture, Forestry and Fisheries from 2013 to 2016, Japan.

**Data Availability Statement:** The data that support the findings of this study are available from the corresponding author upon reasonable request.

**Acknowledgments:** We thank Kazuyuki Inubushi, Chiba University, and Ryusuke Hatano, Hokkaido University for providing valuable comments throughout the study, and Kazuyuki Yagi of the National Agricultural and Food Research Organization (NARO), Institute for Agro-Environmental Science (present affiliation: King Mongkut's University of Technology Thonburi), Osamu Nagata of NARO, Hokkaido National Agricultural Research Center for their helpful advice and support. Furthermore, we thank Seiichi Nishimura of NARO, Hokkaido National Agricultural Research Center, for their suggestions in compiling this paper.

**Conflicts of Interest:** The authors declare no conflict of interest.

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
