# Peer review of "Effect of Paddy-Upland Rotation System on the Net Greenhouse Gas Balance as the Sum of Methane and Nitrous Oxide Emissions and Soil Carbon Storage: A Case in Western Japan"

_agriculture, doi:10.3390/agriculture11010052_

Round 1

Reviewer 1 Report

General comments:

The manuscript addresses methane and nitrous oxide flux over three years comparing the land use decisions of paddy upland rotations with wheat and soybean compared to continuous cultivation of rice in Japan. This is an important topic (GHG flux and alternative land use practices) and clearly appropriate for the journal Agriculture.

I have some reservations about the article. First, it would be useful to put the GHG budgets in terms of yield, and do so in the main text (not in the Supplemental material). Scaling GHG by yield is a very common metric and helps to ascertain if GHG savings are “worth it” from an economic perspective. Yield data need to be included for all crops studied. Secondly, there were substantial differences in the baseline soil physiochemical properties, and while reported, these data were not explored in any meaningful way as they could pertain to the results following the experimental manipulations. Lastly, and my principal concern, is a very poor statistical treatment of otherwise very interesting results. I do not think 2-way ANOVA is appropriate (should be a repeated measures model), and the authors do not provide any justification for using ANOVA since they do not report a test of heteroskedacity, normality, etc. Furthermore, at no point in the results to the authors report test statistics or P-values. Thus, at this point, I cannot recommend the article for publication without a more rigorous treatment of the statistical analysis, but would like to see that issue resolved and a revised manuscript that takes those comments seriously would be considered for publication in Agriculture.

Specific comments:

Methods

-why was there a discrepancy between the sub-plot size for the CP and PU fields? I don’t think this is a red flag, but needs to be explained

  1. 126 on: I’m confused by the gas sampling regime. The authors first state that sampling was weekly, then they state it was two to three times per week, then give some plant growth metrics for when sampling was done? Maybe a table that outlines the actual dates of sampling would be helpful.
  2. 133 on: Again, this is confusing. I think what the authors did was use the described sampling for each of the rotations? However it reads as if a single chamber was covering rice, wheat and soy. More clarity needed here.
  3. 139: I don’t think you mean “stirred”, I’d go with “mixed”
  4. 192: should show the actual calculation/formula used.
  5. 195: what assumption checks were performed? I’d like to see a heterogeneity of variance test at minimum to consider the ANVOA results valid

Figure 1: need to mention in the legend what the colors are, that is easier to interpret than just mentioning the symbols.

Table 3: no interaction terms are shown for year x treatment. This needs to be included. I am also curious to know why the authors used time as a main effect instead of a repeated measures ANOVA model? If samples were taken from the same sub-plots this is more appropriately modeled as a repeated measures design with treatment as the main effect.

  1. 214 on: no report of the F statistics, degrees of freedom and actual P-values is presented. This MUST be included for a proper assessment of the results.
  2. 262 on: see comment above

Discussion

I really like Figure 3, and appreciate Table 4 for putting this into context.

Author Response

December 18, 2020

Dear Reviewer 1,

Thank you very much for your critical and constructive comments on our manuscript on “Effect of paddy-upland rotation system on the net greenhouse gas balance as the sum of methane and nitrous oxide emissions and soil carbon storage: a case in western Japan.” We revised our manuscript based on the editor’s and the reviewers’ comments. Please find our revised manuscript and review once again to consider for publication to consider for publication as a contribution to the special issue on “Sustainable Rice Farming and Greenhouse Gas Emissions” in “Agriculture”. For the details of the revision of individual points, please see the following comments.

< General comments >

  1. Yield data
  • Thank you for your important suggestion. We added a section 3.4 and Table 4 to include the information on yield: “3.4. Yield, quality, and nitrogen uptake of rice, wheat and soybean. The yield, quality, and nitrogen uptake of rice, wheat, and soybean are shown in Table 4. The yield of paddy rice was 692 g m-2 in 2013, 532 g m-2 in 2014, and 580 g m-2 in 2015 in the CR field. In the PU field, it was 552 g m-2 in 2014 and 603 g m-2 in 2015. There were no clear differences in both treatments in the yield of paddy rice both in 2014 and 2015. There were no evident differences in appearance quality and brown rice protein content both in 2014 and 2015, nor in the nitrogen uptake of the above-ground parts (unhulled rice + straw). The yield of wheat in the PU field was 507 g m-2, and the yield of soybean was 505 g m-2, the standard yield [25] was secured. The nitrogen uptake of the above-ground parts (wheat: grain + straw, soybean: grain + stem + pod) was 10.2 g N m-2 for wheat and 33.8 g N m-2 for soybean.” (L331-339, Table 4)
  1. There were substantial differences in the baseline soil physiochemical properties, and while reported, these data were not explored in any meaningful way as they could pertain to the results following the experimental manipulations.
  • Thank you for your important suggestion. We added the following sentences: “In PU field, the values of T-C and T-N were lower than the values in CP, which was considered due to the influence of three rounds of PU rotation until the start of the survey. Detail information on the estimation of temporal change in T-C using Roth C model is described in section 2.4. (L96-99).
  1. (Statistical treatment) my principal concern, is a very poor statistical treatment of otherwise very interesting results. I do not think 2-way ANOVA is appropriate (should be a repeated measures model), and the authors do not provide any justification for using ANOVA since they do not report a test of heteroskedacity, normality, etc. Furthermore, at no point in the results to the authors report test statistics or P-values. Thus, at this point, I cannot recommend the article for publication without a more rigorous treatment of the statistical analysis, but would like to see that issue resolved and a revised manuscript that takes those comments seriously would be suitable for publication in Agriculture.
  • Thank you for your important suggestion. We would like to explain the statistical treatment more clearly. Basically, we set up 3 sub-plots in each plot (CP and PU) and field data on GHG emissions and yield etc. were obtained (L110-112). We regard these 3 measurements as 3 replications, judging from the fact this is a large-scale field experiment. Based on this background, using the three-year survey data, differences in GHG emissions in two treatments were compared by two-way analysis of variance (ANOVA, treatment and year) (L206-208). Interaction term, i.e. treatment × year, is also investigated based on your recommendation. The overall results can be seen in revised Table 3 and description on statistical significance is added to the manuscript (L206-208, 231, 240-245, 249-250, 285-286, 293, 299-300, 306-307, 314-315)
  •  

< Specific comments >

  1. Why was there a discrepancy between the sub-plot size for the CP and PU fields?
  • Thank you for your inquiry. We added the following sentence: “The size of sub-plot was different between CP and PU reflecting the planting style because wheat and soybean were cultivated on ridges, which had longer width than the width of the rows of rice.” (L112-114)
  1. L. 126 on: I’m confused by the gas sampling regime. The authors first state that sampling was weekly, then they state it was two to three times per week, then give some plant growth metics for when sampling was done? Maybe a table that outlines the actual dates of sampling would be helpful.
  • Thank you for your important suggestion. We corrected the corresponding sentences as follows: “Gas was basically collected once a week between 9:00 AM and 12:00 AM. Specially, the frequency of measurement was about two to three times per week for the puddling and transplanting stages, before and after mid-summer and pre-harvest drainages in the paddy rice cultivation period, and immediately after the fertilization of wheat and soybean. In contrast, the measurement frequency was biweekly only from January to March in the post-soybean fallow period and the fallow period of paddy.” (L127-132)
  1. L. 133 on: Again, this is confusing. I think what the authors did was use the described sampling for each of the rotations? However it reads as if a single chamber was covering rice, wheat and soy. More clarity needed here.
  • Thank you for your important suggestion. We added the following phrase: “measurement of each sub plot.” (L133-134)
  1. L. 139: I don’t think you mean “stirred”, I’d go with “mixed”.
  • Thank you for your important suggestion. I revised the text as suggested. (L139).
  1. L. 192: should show the actual calculation/formula used.
  • Thank you for your important suggestion. We added the following sentence: “Annual CO2eq data of both CH4 and N2O emissions were used as the data of GHG emissions, and the amount of soil carbon stored annually in the CP and PU fields were regarded as soil carbon storage and converted to the CO2eq emissions. The calculation formula is as follows.If the value was positive, it suggested a net emission, and if the value was negative, it suggested a net absorption. The mitigation effect was evaluated by subtracted the CP value from the PU value. (L199-204)
  • Net GHG balance (Mg CO2eq ha-1 year-1) = GHG emissions+CO2eq emissions”
  1. L. 195: what assumption checks were performed? I’d like to see a heterogeneity of variance test at minimum to consider the ANVOA results valid.
  • Thank you for your important suggestion. The validity of ANOVA can be estimated by the comparison of the average and standard deviation of the data in Table 3. Validity was also confirmed from the results of the statistical analysis.
  1. Figure 1: need to mention in the legend what the colors are, that is easier to interpret than just mentioning the symbols.
  • We added the following sentence. “×(Blue) :CP, â—‹(Red) :PU” (L218)
  1. Table 3: no interaction terms are shown for year x treatment. This needs to be included. I am also curious to know why the authors used time as a main effect instead of a repeated measures ANOVA model? If samples were taken from the same sub-plots this is more appropriately modeled as a repeated measures design with treatment as the main effect.
  • Thank you for your important suggestion. As explained above, we set up 3 sub-plots in each plot (CP and PU) and field data on GHG emissions and yield etc. were obtained (L110-112). We regard these 3 measurements as 3 replications, judging from the fact this is a large-scale field experiment. Based on this background, using the three-year survey data, differences in GHG emissions in two treatments were compared by two-way analysis of variance (ANOVA, treatment and year) (L206-208). We added the results of interaction terms in Table 3 (Table3)
  1. L. 214 on: no report of the F statistics, degrees of freedom and actual P-values is presented. This MUST be included for a proper assessment of the results.
  2. Thank you for your important suggestion. We added the following sentence: “T-test was performed for CH4 and N2O flux” (L207-208) “In PU, the CH4 flux in rice cultivation was significantly lower than that in CP, during the period from one month after water injection (May 28) to the pre-harvest drainage (August 11) in the first year of paddy rice cultivation after upland crops, except June 19 and June 23 (p<0.05). Also in the second year of paddy rice cultivation after upland crops, it was significantly lower than that in CP, during the period from the end of midsummer drainage (June 25) to the pre-harvest drainage (August 27), except for August 4, August 10, and August 24 (p<0.05).” (L 240-245) “which were significantly higher than those in the CP field (p<0.05) ” (L249-250)
  3. L. 262 on: see comment above

Thank you for your important suggestion. We added the following sentence and revised Table3. “T-test was performed for CH4 and N2O flux, CH4, N2O, and CO2eq emissions in each year.” (L207-208) “The CH4 cumulative emissions in the PU field were significantly lower than those in the CP field in each three years (p<0.05).” (L285-286) “significantly higher than those in the CP field in the first year (p<0.05).” (L293) “The GWP of CH4 cumulative emissions in the PU field were significantly lower than those in the CP field in each three years (p<0.05).” (L299-300) “The GWP of N2O cumulative emissions in the PU field were significantly higher than those in the CP field in the first year (p<0.05).” (L306-307) “The CO2eq cumulative emissions in the PU field were significantly lower than those in the CP field in each three years (p<0.05).” (L314-315)

Reviewer 2 Report

This study conducted a three-year field experiment and evaluated two cropping systems (PU rotation vs. CP) regarding their GHG emissions. The authors found that PU rotation can greatly reduce the overall greenhouse effect compared to CP. The manuscript is well organized and presented with sufficient measured data to support the conclusion. The reviewer found no major issues and following are some revision suggestions:

  1. The authors should briefly state in the introduction why CO2 emissions cannot be actually measured in crop fields and must be estimated with models. In addition, the authors should briefly describe why the RothC model is superior to other models available with similar functions.
  2. Line 23: A typo: CR field should be CP field. 
  3. Line 145: Are these GWP conversion factors based on the 100-year GWP? The reviewer found the 100-year GWP conversion factors of CH4 and N2O to be 25 and 298, respectively, on the USEPA website. Please verify the values using reliable resources.
  4. Line 189: Please define RMSE.
  5. Line 382: It is hard to understand the sentence "reflecting higher temperature and lower soil moisture is low".

Author Response

Thank you very much for your critical and constructive comments on our manuscript on “Effect of paddy-upland rotation system on the net greenhouse gas balance as the sum of methane and nitrous oxide emissions and soil carbon storage: a case in western Japan.” We revised our manuscript based on the editor’s and the reviewers’ comments. Please find our revised manuscript and review once again to consider for publication to consider for publication as a contribution to the special issue on “Sustainable Rice Farming and Greenhouse Gas Emissions” in “Agriculture”. For the details of the revision of individual points, please see the following comments.

  1. The authors should briefly state in the introduction why CO2 emissions cannot be actually measured in crop fields and must be estimated with models. In addition, the authors should briefly describe why the RothC model is superior to other models available with similar functions.
  • Thank you for your important suggestion. We added the following sentence to support our methodology: “As the change in soil carbon is generally slow and there may be large spatial variations, it takes a long time to detect the change by actual measurement. In this study, therefore, changes in soil carbon content were calculated using the Roth C model, which had been verified to be simple and accurate [34, 35, 36].” (L169-171).
  1. Line 23: A typo: CR field should be CP field.
  • Thank you for finding my careless mistake. I revised the text as suggested. (L23)
  1. Line 145: Are these GWP conversion factors based on the 100-year GWP? The reviewer found the 100-year GWP conversion factors of CH4 and N2O to be 25 and 298, respectively, on the USEPA website. Please verify the values using reliable resources.
  • Your suggested data, the 100-year GWP conversion factors of CH4 and N2O to be 25 and 298, are based on the IPCC Fourth Assessment Report (AR4). As we used the latest 100-year GWP conversion factors of CH4 and N2O to be 28 and 265 in AR5, we suppose these updated factors would be more suitable to use in this manuscript. We hope you agree with our opinion. (L145-146).
  1. Line 189: Please define RMSE.
  • Thank you for your important suggestion. We added the following sentence to respond to your suggestion: “RMSE (The root mean square error) were calculated from the differences between predicted values by the RothC model and the measured values for each field. The RMSEs in the PU and CP fields were 2.64 and 2.39, respectively, suggesting similar and relatively small error in both fields.” (L191-194).
  1. Line 382: It is hard to understand the sentence "reflecting higher temperature and lower soil moisture is low".
  • Thank you for your important suggestion. We revised the sentence to clarify the meaning: “This would be because decomposition of organic matter in Thailand proceeded much faster than the other two cases due to more oxidative state, reflecting higher temperature and lower precipitation.” (L416-418)Thank you in advance for reviewing our manuscript once again.
  •  
  • Sincerely,Hiroyuki Hasukawa
  • Agricultural Technology Promotion Center of Shiga Prefecture
  •  
  •  
  •  

Reviewer 3 Report

The manuscript entitled “Effect of paddy-upland rotation system on the net greenhouse gas balance as the sum of methane and nitrous oxide emissions and soil carbon storage: a case in western Japan” has been critically reviewed. The manuscript is having high scientific merits as the global warming is one of the biggest threats of 21st century. Anthropogenic methane and nitrous oxide emission play significant role in global climate change and their management is essential for combating global warming.  In my opinion, this manuscript should make some major revisions.

The point wise comments of the manuscript are enlisted below:

Line 14, 34, 35, so on: greenhouse gas- it will be greenhouse gases, kindly check in further also in manuscript

Line no 27: replace paddy-upland by PU

Line no 35: It should be reference [1] not [8]. References are to be re-arranged as per Journal guidelines.    

Line no: replace reference no 43 by relevant reference (suggested reference for citation https://doi.org/10.1038/srep44928; https://doi.org/10.1080/17583004.2020.1752061)

Line no: Since about 90% of paddy rice is produced in Asia [5]. Kindly updated reference year 2015, by 2020.

Line no 41: Replace reference no 11 by https://doi.org/10.1016/j.bcab.2019.101266; https://doi.org/10.1016/j.agee.2016.05.023;

Line no 44: Please cite the following https://doi.org/10.1016/j.scitotenv.2016.07.182 published paper to updated manuscript.

Line no 44: remove word paddy and write scientific name of rice also.

Line 52-54: Kumagai and Konno (1998) [12], Nishimura et al. (2008) [16], Nishimura et al. (2011) [17], and Shiono et al. (2014) [27] reported the effect of PU rotation system on the reduction of methane (CH4) emissions. Replace by  Kumagai and Konno[12], Nishimura et al.[16], Nishimura et al. [17], and Shiono et al. [27] reported the effect of PU rotation system on the reduction of CH4 emissions

Line no 58: Delete-(Chu et al., 2004; Nishimura et al., 2011; Shiono et al., 2014).

Line no 61: Delete-Sumida et al. (2005)

Line no 62-Delete-(2014)

Line no 35, 65, 69, 72, 77: replace greenhouse gas by GHG (kindly check other section of manuscript and update accordingly)

Paddy rice and rice ??? S1, S2 and S3????

Line

Table 1- please use either paddy or rice word in place of paddy rice. And follow uniform in whole complete draft, kindly check and update accordingly.   

Line no 127- references is missing of closed-chamber method??

Line no: 149 to 159 remove this from section 2.2 and add it in to the section 2.3.

Section 2.5. Kindly elaborate it and support with relevant reference.

Please check super or sub fix in word in following line no- 234, table 3, fig. 1, redox potential (Eh) in complete manuscript,

If possible kindly redraw Fig 1 for increasing visibility

Line no- 239- The cumulative precipitation in 2013, 2014 and 2015 was 1,520 mm, 1,583 mm and 1,800 mm respectively. Complete sentence by adding word respectively.

Line no 240, add word respectively in the sentence.

Line no 240, kindly mention its mean or max or min air temperature???

 Title: If possible make it title short.

Author Response

Dear Reviewer 3,

Thank you very much for your critical and constructive comments on our manuscript on “Effect of paddy-upland rotation system on the net greenhouse gas balance as the sum of methane and nitrous oxide emissions and soil carbon storage: a case in western Japan.” We revised our manuscript based on the editor’s and the reviewers’ comments. Please find our revised manuscript and review once again to consider for publication to consider for publication as a contribution to the special issue on “Sustainable Rice Farming and Greenhouse Gas Emissions” in “Agriculture”. For the details of the revision of individual points, please see the following comments.

  1. Line 14, 34, 35, so on: greenhouse gas- it will be greenhouse gases, kindly check in further also in manuscript.
  • Thank you for finding my careless mistake. I revised the text as suggested. (L14, 34, 38)
  1. Line no 27: replace paddy-upland by PU.
  • Thank you for finding my careless mistake. I revised the text as suggested. (L27).
  1. Line no 35: It should be reference [1] not [8]. References are to be re-arranged as per Journal guidelines.
  • The order was corrected as suggested.
  1. Line no: replace reference no 43 by relevant reference (suggested reference for citation https://doi.org/10.1038/srep44928; https://doi.org/10.1080/17583004.2020.1752061).
  • Thank you for your important suggestion. I replaced and added the references. (L41, 495-498)
  1. Line no: Since about 90% of paddy rice is produced in Asia [5]. Kindly updated reference year 2015, by 2020.
  • Thank you for your important suggestion. I revised the text as suggested. (L499).
  1. Line no 41:
  2. Replace reference no 11 by https://doi.org/10.1016/j.bcab.2019.101266; https://doi.org/10.1016/j.agee.2016.05.023;.
  • Thank you for your important suggestion. Since this sentence describes on methane, reference No.11 (No. changed 6) was left and No.7 was added. (L41, 505-507)
  1. Line no 44:
  2. Please cite the following https://doi.org/10.1016/j.scitotenv.2016.07.182 published paper to updated manuscript.
  • Thank you for your important suggestion. I added the reference. (L44, 522-524)
  1. Line no 44: remove word paddy and write scientific name of rice also.
  • There are upland rice and paddy rice for rice production, but this study is a survey of paddy rice, and I would like to describe it as paddy rice. We added the scientific name of rice: “In this context, paddy rice (Oryza sativa) ” (L44)
  1. Line 52-54: Kumagai and Konno (1998) [12], Nishimura et al. (2008) [16], Nishimura et al. (2011) [17], and Shiono et al. (2014) [27] reported the effect of PU rotation system on the reduction of methane (CH4) emissions. Replace by Kumagai and Konno[12], Nishimura et al.[16], Nishimura et al. [17], and Shiono et al. [27] reported the effect of PU rotation system on the reduction of CH4 emissions.
  • Thank you for finding my careless mistake. I revised the text as suggested. (L52)
  1. Line no 58: Delete-(Chu et al., 2004; Nishimura et al., 2011; Shiono et al., 2014).
  • Thank you for finding my careless mistake. I revised the text as suggested. (L58)
  1. Line no 61: Delete-Sumida et al. (2005).
  • Thank you for finding my careless mistake. I revised the text as suggested. (L60)
  1. Line no 62-Delete-(2014).
  • Thank you for finding my careless mistake. I revised the text as suggested. (L61)
  1. Line no 35, 65, 69, 72, 77: replace greenhouse gas by GHG (kindly check other section of manuscript and update accordingly).
  • The order was corrected as suggested. (L35, 65, 66, 68, 71, 73, 75, 195, 196, 329, 330, 334, 341, 342, 433)
  1. Paddy rice and rice ??? S1, S2 and S3????
  • Thank you for your important suggestion. We used paddy rice word. (S1, S3)
  • In addition, the Table S3 has been changed to Table4 according to the opinion of another reviewer. (Table4)
  1. Table 1- please use either paddy or rice word in place of paddy rice. And follow uniform in whole complete draft, kindly check and update accordingly.
  • Thank you for your important suggestion. We want to describe it as paddy rice because we want to specify it including cropping management without confusing the reader. (Table1)
  1. Line no 127- references is missing of closed-chamber method??
  • Thank you for your important suggestion. I added the references. (L127).
  1. Line no: 149 to 159 remove this from section 2.2 and add it in to the section 2.3.
  • The order was corrected as suggested. We added the following sentence: “ Measurement of selected field properties” (L149,157-167)
  1. Section 2.5. Kindly elaborate it and support with relevant reference.
  • Thank you for your important suggestion. We added the following sentence to respond to your suggestion: “Annual CO2eq data of both CH4 and N2O emissions were used as the data of GHG emissions, and the amount of soil carbon stored annually in the CP and PU fields were regarded as soil carbon storage and converted to the CO2eq emissions. The calculation formula is as follows.If the value was positive, it suggested a net emission, and if the value was negative, it suggested a net absorption. The mitigation effect was evaluated by subtracted the CP value from the PU value.” (L197-203)
  • Net GHG balance (Mg CO2eq ha-1 year-1) = GHG emissions+CO2eq emissions
  1. Please check super or sub fix in word in following line no- 234, table 3, fig. 1, redox potential (Eh) in complete manuscript.
  • The order was corrected as suggested. (L251, Table 3, Figure 1)
  1. If possible kindly redraw Fig 1 for increasing visibility.
  • As the most important points, i.e. the period when CP and PU showed different trend, is clear from this figure, we would like to keep this figure as it is. (Figure 1)
  1. Line no- 239- The cumulative precipitation in 2013, 2014 and 2015 was 1,520 mm, 1,583 mm and 1,800 mm respectively. Complete sentence by adding word respectively.
  • Thank you for your important suggestion. We added the word “respectively” as suggested. (L257)
  1. Line no 240, add word respectively in the sentence.
  • Thank you for your important suggestion. We added the word as suggested. (L258)
  1. Line no 240, kindly mention its mean or max or min air temperature???
  • Thank you for your important suggestion. We added the following word to clarify the sentence: “The mean air temperature …” (L257)
  1. Title: If possible make it title short.
  • Thank you for your important suggestion. We understand that this is a bit long title, but we prefer to clearly describe the meaning of “the net greenhouse gas balance” as “the sum of methane and N2O emissions and soil carbon storage” in the title. We hope you understand our decision.Thank you in advance for reviewing our manuscript once again.
  •  
  • Sincerely,Hiroyuki Hasukawa
  • Agricultural Technology Promotion Center of Shiga Prefecture
  •  
  •  
  •  

Round 2

Reviewer 1 Report

I still have reservations about this manuscript based on the lack of rigor in the statistical treatment. No attempt was made at a formal variance structure analysis (my previous point #6), and test statistics are still missing from the text (point #9). 

Author Response

December 27, 2020

Dear Reviewer 1,

Thank you very much for your critical and constructive comments on our manuscript on Effect of paddy-upland rotation system on the net greenhouse gas balance as the sum of methane and nitrous oxide emissions and soil carbon storage: a case in western Japan.” We revised our manuscript based on the reviewers’ second comments. Please find our revised manuscript and review once again to consider for publication to consider for publication as a contribution to the special issue on “Sustainable Rice Farming and Greenhouse Gas Emissions” in “Agriculture”. For the details of the revision of individual points, please see the attached file.

Reviewer 3 Report

Manuscript is extensively revised.

Kindly look for one or two typographical error mention below:   

Please check for typographical error (delete reference year, for example in line no 65 -Takakai et al. (2017) [22] to be written as Takakai et al. [22]). Also update similarly in line no 151, 180, 141, 219, 363, 372, 373, 377, 389, 390, 397, 399, 400, 405, 409, 410, 415, 419, 436, 439, 443,

Table 1 and Fig. 1- Kindly use word paddy or rice in place of paddy rice

Author Response

December 27, 2020

Dear Reviewer 3,

Thank you very much for your critical and constructive comments on our manuscript on “Effect of paddy-upland rotation system on the net greenhouse gas balance as the sum of methane and nitrous oxide emissions and soil carbon storage: a case in western Japan.” We revised our manuscript based on the reviewers’ second comments. Please find our revised manuscript and review once again to consider for publication to consider for publication as a contribution to the special issue on “Sustainable Rice Farming and Greenhouse Gas Emissions” in “Agriculture”. For the details of the revision of individual points, please see the following comments.

Kindly look for one or two typographical error mention below:  

  1. Please check for typographical error (delete reference year, for example in line no 65 -Takakai et al. (2017) [22] to be written as Takakai et al. [22]). Also update similarly in line no 151, 180, 141, 219, 363, 372, 373, 377, 389, 390, 397, 399, 400, 405, 409, 410, 415, 419, 436, 439, 443,
  • Thank you for finding my careless mistake. I revised the text as suggested.
  1. Table 1 and Fig. 1- Kindly use word paddy or rice in place of paddy rice
  • Thank you for your important suggestion. However, we want to describe it as “paddy rice” to indicate “rice cultivated under paddy (waterlogging management) so as not to be confused with “upland rice”. We really hope you kindly understand our intention. (Table1, Fig1)

Thank you in advance for reviewing our manuscript once again.

Sincerely,

Hiroyuki Hasukawa

Agricultural Technology Promotion Center of Shiga Prefecture

Round 3

Reviewer 1 Report

no more comments